2018–2022 Southern Resident killer whale presence in the Salish Sea: continued shifts in habitat usage

Shields Monika W. monika@orcabehaviorinstitute.org
Orca Behavior Institute , Friday Harbor , WA , United States of America
Vasapollo Claudio
Electronic publication date: 2023 Jul 12
Publication date: 2023
Volume: 11
Electronic Location ID: e15635
Received 2023 Feb 20; Accepted 2023 Jun 5
Copyright: ©2023 Shields
Copyright year: 2023
Copyright holder: Shields
License: This is an open access article distributed under the terms of the Creative Commons Attribution License, which permits unrestricted use, distribution, reproduction and adaptation in any medium and for any purpose provided that it is properly attributed. For attribution, the original author(s), title, publication source (PeerJ) and either DOI or URL of the article must be cited.
License URL: https://creativecommons.org/licenses/by/4.0/

Keywords: Orca, Killer whale, Endangered species, Salish Sea, Wildlife tracking, Orcinus orca, Presence-absence, Habitat usage

Funding: The author received no funding for this work.

==============================
The fish-eating Southern Resident killer whales (Orcinus orca) of the northeastern Pacific are listed as Endangered in both the USA and Canada. The inland waters of Washington State and British Columbia, a region known as the Salish Sea, are designated as Southern Resident critical habitat by both countries. The whales have historically had regular monthly presence in the Salish Sea, with peak abundance occurring from May through September. In recent years, at least partially in response to shifting prey abundance, habitat usage by the Southern Residents has changed. As conservation measures aim to provide the best possible protection for the whales in their hopeful recovery, it is key that policies are based both on historic trends and current data. To this aim, our study shares 2018–2022 daily occurrence data to build upon and compare to previously published whale presence numbers and to demonstrate more recent habitat shifts. Based on reports from an extensive network of community scientists as well as online streaming hydrophones, every Southern Resident occurrence was confirmed either visually or acoustically. Documented here are the first-ever total absence of the Southern Residents in the Salish Sea in the months of May, June, and August, as well as their continued overall declining presence in the spring and summer, while fall and winter presence remains relatively high. It is key that management efforts consider these shifting presence patterns when setting both seasonal and regional protection measures aimed at supporting population recovery.

Introduction

For marine mammals, marine protected areas (MPAs) are an increasingly common tool utilized in conservation (Hoyt, 2017). This broad term is applied to a wide variety of marine preserves, parks, or sanctuaries, each with differing levels of conservation measures in place ranging from limitations on actitivities such as fishing to slow-transit zones or full vessel exclusion areas. Historically, many MPAs for marine mammals have been too small and/or too weakly regulated and enforced to have the intended conservation impact; one step identified as necessary for making MPAs more effective is the regular incorporation of new information to adapt to quickly changing ecosystems (Hoyt, 2017).

Following an Endangered listing in either Canada or the US, a critical habitat for the listed species is designated. Critical habitat is defined as the geographic area occupied by the species that contains the physical and/or biological features essential to the conservation of that species (Criteria for Designating Critical Habitat, 2002). These types of habitat protections are essential when supporting endangered species recovery but may also lead to a static view on habitat usage that can result in ineffective management efforts that only have a marginal positive impact on conservation (Hiers et al., 2016). With ecosystems changing rapidly in the modern world due to a wide variety of anthropogenic impacts, it is important to monitor real-world variation in habitat usage over time so that spatial and temporal related recovery efforts are updated to reflect current conditions rather than historic trends. As stated in Hiers et al. (2016), “The broader policy frameworks must evolve to allow more rapid adaptation to changing circumstances and quicker adoption of better information as it becomes available”. One role of scientists in this process is to make that data available in a timely manner.

The North Atlantic right whale (Eubalaena glacialis) provides one example of where habitat usage shifted in response to prey availability, prompting a timely response in management efforts to alter MPAs. Following their Endangered listing, critical habitat for right whales was designated in both the US and Canada and these designations informed the timing and placement of vessel slow-down zones, vessel route changes, and fishing restrictions designed to reduce fatalities and thus aid in population recovery. In 2017, the right whales shifted their feeding grounds further north, outside of the protected habitats. Within weeks, emergency protection measures were implemented to reduce the risk of fatal vessel strikes in this newly utilized region (Meyer-Gutbrod, Greene & Davies, 2018).

The salmonid-eating Southern Resident killer whales (Orcinus orca) of the northeast Pacific are considered one of the most well-known wild cetacean populations in the world. They have been the focus of ongoing population monitoring since the early 1970s (Bigg et al., 1987), with their small population size and coastal habits making them amenable to close observation impossible with many other marine species. Made up of J-, K-, and L-Pods and totaling 73 individuals as of the end of 2022 (Center for Whale Research, 2023), each individual’s personal and familial life history is known in detail (Ford, Ellis & Balcomb, 1994).

After an approximately 20% population decline in the late 1990s and early 2000s, the Southern Resident killer whales (SRKW) were listed as Endangered in Canada in 2001 (Fisheries and Oceans Canada, 2018) and the USA in 2005 (National Marine Fisheries Service, 2008), with identified risk factors inhibiting their recovery including reduced prey availability, toxic contamination, and vessel effects. Their endangered status has prompted a wide array of additional studies focused on topics such as their diet (Hanson et al., 2010; Hanson et al., 2021), hormone levels (Ayres et al., 2012), body condition (Stewart et al., 2021), and responses to vessel disturbance (Houghton et al., 2015; Holt et al., 2021).

The initial critical habitat designation for SRKW in the United States occurred in 2008 and was defined as the inland waters of Washington State, a region just over 2,500 square miles, split into three areas: (1) the US waters around the San Juan Islands including Haro Strait (a region defined as the “Core Summer Area”), (2) all of Puget Sound, and (3) the US waters of the Strait of Juan de Fuca (National Marine Fisheries Service, 2008). In 2021, the designated critical habitat was revised to include an additional 15,910 square miles along the outer coastal regions of Washington, Oregon, and California (National Marine Fisheries Service, 2021). The initial critical habitat designation in Canada, which also occurred in 2008, was identified as the transboundary waters of southern British Columbia, including the southern Strait of Georgia, Haro Strait, and the Strait of Juan de Fuca. This region was recognized as being of specific importance as a foraging area to all three pods from June through October. In 2018, the Canadian critical habitat expanded to include waters on the continental shelf off southwestern Vancouver Island (Fisheries and Oceans Canada, 2018).

The transboundary inland waters of Washington and British Columbia are known as the Salish Sea, and the designation of these waters as critical habitat reflected the historic summer usage of this area by SRKW (Ford, Ellis & Balcomb, 2000). Commonly referred to as the SRKW core summer habitat, all three pods regularly utilized the central Salish Sea from May through September, shifting further south to visit Puget Sound in the fall. K- and L-Pods would spend most of their time on the outer coast in the late fall through early spring, while J-Pod remained in the Salish Sea for much of the year. Nearly 40 years of tracking from 1976–2014 reported that SRKW presence was confirmed in the Salish Sea an average of 193 days per year (Olson et al., 2018). A study spanning 1996–2001 found the SRKW were present in the central Salish Sea an average of 79.3% of the days in May through September (Hauser et al., 2007).

Diet studies have demonstrated that the seasonal movements of SRKW correspond to changes in prey availability. While SRKW selectively feed on the comparatively rare Chinook and chum salmon over the more abundant pink and sockeye salmon, the historic occurrence patterns of the whales reflect the run timings of their preferred prey (Ford & Ellis, 2006). While Chinook salmon make up the majority of the year-round diet of the Southern Residents, there is some seasonal variation, with other salmonid species making up a greater proportion of the diet outside the summer season. The specific Chinook stocks the whales rely on also vary throughout the year. Fraser River Chinook salmon have traditionally dominated the summer diet (Hanson et al., 2010), explaining their extended presence in the central Salish during the months of May through September. In the fall, there is a shift to a much greater proportion of chum salmon in the SRKW diet (Hanson et al., 2021), particularly while whales are in inland waters and corresponding to their visits to Puget Sound. With Chinook and chum salmon stocks varying widely in their numbers from year to year (Losee, Kendall & Dufualt, 2019) and Chinook declining coast-wide overall (Dorner, Catalano & Peterman, 2017), the SRKW must adapt to changing prey availability, and will have to shift their habitat usage as a result.

A few changes in habitat usage by SRKW over the last few decades have been noted in the literature. Beginning in 1999, Ks and Ls became more frequent fall visitors to Puget Sound (Olson et al., 2018), and beginning in 2005 all three pods’ presence in the Salish Sea declined during the spring months of April through June in correlation to declining returns of spring-run Chinook to the Fraser River (Shields, Lindell & Woodruff, 2018). An additional study corroborated these results, finding that from 2001–2017, peak occurrence of SRKW within the summer core habitat has shifted to be 1–5 days later per year on average, resulting in a 17–85 day later peak occurrence shift across the 17 year time period (Ettinger et al., 2022). This was also found to be consistent with the declining spring Fraser River Chinook runs.

Management efforts in both the US and Canada have included temporal and geographic protection areas in an attempt to mitigate vessel effects and increase prey availability for the SRKW. In both countries, these include specific regulations that overlap regions of the central Salish Sea and the “core summer months” based on historic sightings patterns (Transport Canada, 2023; Washington State Legislature, 2019). The aim of this study is to provide an update on the seasonal and annual usage of the Salish Sea by each of the Southern Resident pods and the SRKW population as a whole by assessing confirmed presence over the five year period from 2018–2022, with the goal of providing more recent habitat usage data to inform protection measures.

Materials & Methods

Whale sightings and identification

From January 1, 2018 to December 31, 2022, orca sighting reports throughout the Salish Sea were tracked daily from a variety of sources. These sources included research encounters by the Orca Behavior Institute; reports from the Pacific Whale Watch Association shared via their private social media sightings page and proprietary sightings app; public reports shared via Orca Network; public encounter summaries from the Center for Whale Research; public sightings reported via social media, often through regional community sightings pages; the Salish Sea Orcasound hydrophone network; and community member reports submitted directly to the Orca Behavior Institute. All SRKW reports were verified visually and/or acoustically by the author, utilizing photos, videos, and/or audio from the observers and referencing established SRKW photo ID catalogues from the Center for Whale Research and the (Ford, 1987) acoustic call catalogue for killer whales in British Columbia. This type of visual (e.g., Fearnbach et al., 2011) and acoustic (e.g., Riera et al., 2019) identification are both well established methodologies for SRKW.

Initial and final sighting locations were noted for each day along with travel routes, and geographic region within the Salish Sea was also noted. For the purpose of this study, the Salish Sea was defined as all the inland waters east of Otter Point, BC including Puget Sound and the Strait of Georgia. Geographic regions were defined as: Northern Salish Sea (NSS), north of a line connecting Nanaimo and Vancouver, BC; Puget Sound (PS), south of the Port Townsend-Coupeville ferry lanes and east of the Deception Pass Bridge; and Central Salish Sea (CSS) for all the waters in between (Fig. 1). Seasonal presence was noted as winter (Jan–Mar), spring (Apr–Jun), summer (Jul–Sep), and fall (Oct–Dec).

Figure 1 Map of the Salish Sea study area.

SRKW sightings were tracked for all the Salish Sea waters east of Otter Point, British Columbia. The region was further divided into the Northern Salish Sea (NSS, teal), Central Salish Sea (CSS, pink), and Puget Sound (PS, yellow).

In addition to confirming the Southern Resident ecotype (as opposed to the sympatric population of Bigg’s killer whales), pod(s) present (J, K, or L) were also confirmed based on the media provided. While it was impossible to identify all individuals present through this method, the presence of a single individual from a given pod was deemed sufficient to confirm presence of members of that pod on a given day. The only exception was L87, an L-Pod adult male who routinely traveled with J-Pod from prior to 2018 until early 2020. Since L87 has been documented traveling long-term with each of the three pods, his presence was not considered sufficient to identify the presence of a specific pod.

Any data set making use of a variety of sightings sources is subject to observer bias that cannot be tested or controlled for. In this data set, observer effort was greater in the CSS and PS regions than the NSS. Western Strait of Juan de Fuca reports were sporadic enough that for this report the waters west of Otter Point, BC were excluded; the whales are also likely under-reported in the north-central Strait of Georgia which is a less populated area with more open waterways. Observer effort was also higher during spring, summer, and fall months when daylight hours and weather conditions allow for more hours of optimal whale viewing or searching. The increasing popularity of social media platforms to track and share whale reports has also led to more observer effort over time.

In 2020, due to the COVID-19 pandemic, Washington State was under a stay-at-home order from March 18 to May 31, while Transport Canada issued a suspension of all recreational tourism from April 6 until June 30. Coastal homeowners, essential boat traffic (such as ferries), and listeners of Salish Sea hydrophone networks still reported whale detections during this time. Given the tendency of SRKW to be highly vocal, in large groups, and very surface active, we do not believe they went undetected in the Salish Sea during this time, though it is important to note this dip in observer effort over these months, especially since there were no confirmed reports of SRKW in May 2020 (though they were also absent in May of 2019 and 2021).

To account for multiple reports of the same whales on the same day and following previously established methodology (Olson et al., 2018), the metric of “whale day” was used. A whale day is defined as a day of confirmed presence of SRKW (and, similarly, of a specific pod), regardless of the number of reports received on the given day. Despite these acknowledged biases, the cumulative tracking of SRKW in inland waters across these various sightings sources is considered robust, with the whales’ presence unlikely to be missed throughout most of the region during most of the year (Olson et al., 2018; Hauser, 2006).

In an additional effort to check for under-reporting of SRKW, for 2022 only “speculated whale days” were also tracked. A speculated whale day was defined as a day where, given SRKW average travel speeds and known travel routes, they were likely present in the Salish Sea even though they were not reported. For example, on the morning of January 9, 2022, J-Pod was reported in Swanson Channel near Pender Island, BC. This is approximately 100 miles from the entrance to the Salish Sea, so while there were no reports of them on January 8, January 8 was tallied as a speculated whale day since J-Pod must have been entering the Salish Sea via the Strait of Juan de Fuca on that day. Another example of a speculated whale day is January 21. J-Pod was reported going north in Haro Strait off San Juan Island, WA on January 20 and coming south down Boundary Pass near Saturna Island, BC on January 22. While there were no reports of them on January 21, January 21 was counted as a speculated whale day as J-Pod was presumably undetected in the Strait of Georgia.

Spring and summer 2018–2022 SRKW presence was compared to historic sightings records from April through September 2004–2017. These data, which were published in (Shields, Lindell & Woodruff, 2018), were compiled and confirmed in a similar manner as above, with the primary sources being Orca Network and the Pacific Whale Watch Association.

Statistical analysis

Data were imported into R Studio RDE (R Studio Team, 2023) and analyzed with R 4.2.3 (R Core Team, 2023). The count data of number of whale days per month from 2018–2022 was fit to a negative binomial model using the glm.nb() function in the MASS package as follows: glm.nbDays∼Season∗Area+Year

A negative binomial model was chosen over the Poisson due to the overdispersion of the data. An interaction plot of estimated marginal means was created using the emmip() function in the emmeans package.

Monthly whale presence/absence data from April through September for 2004–2022 was fit to a generalized additive model (GAM) using the gam() function in the mgcv package as follows: gamcbindDaysPresent,DaysAbsent∼snMonth,k=5+sYear,k=5+sYearFactor,bs=”re”,family=”quasibinomial”.

The count data for whale presence each month is the result of n binomial trials where n = the number of days in the month. Due to overdispersion of the data, a quasibinomial family was chosen. The model included smoothing parameters to allow for non-linear trends across both months and years, with a year-level random effect to account for additional environmental factors outside of the trend. The results were visualized with the plot() function.

Results

Whale sightings summary

After removing duplicate reports of the same group of whales on the same day, there were 732 unique SRKW sightings over the five-year study period, representing 647 whale days and confirmed SRKW presence in the Salish Sea for 35.5% of the five-year study period. Note that some days there were multiple groups of SRKW present in different locations (defined as >20 miles apart), which is why the number of sightings is greater than the number of whale days. Presence ranged from a low of 103 days in 2021 to a high of 167 days in 2022, with an average of 129.4 days across the five years. J-Pod was present 592 days, K-Pod 190 days, and L-Pod 170 days. As in historic trends, J-Pod was present nearly every month of the year, while K- and L-Pods were generally absent from late winter through early summer (Table 1).

Combining sightings from all pods (n = 732), the most sightings occurred in the fall (260, 35.5%) and the summer (258, 35.2%) followed by the winter (131, 17.9%) and the spring (83, 11.3%) (Fig. 2). Looking at seasonal presence by region, the greatest number of sightings occurred in the Central Salish Sea for the winter (77.1% of total winter sightings), spring (86.8%), and summer (90.3%), while in the fall most sightings were in Puget Sound (51.2% of fall sightings). No Puget Sound sightings were recorded in the months of February or March or in May through August. The Northern Salish Sea sightings ranged from 3.6% to 12.2% of seasonal totals, with higher percentages of sightings occurring in winter and spring. No Northern Salish Sea sightings were documented in the months of May, June, or August. Figure 3 shows a Salish Sea map of all 2018–2022 SRKW sightings by season.

Looking at the 151 days in the May through September period (what was formerly considered the core summer season), SRKW presence varied from 29 days in 2021 to 71 days in 2018, with an average seasonal presence of 32.6% of the days.

In 2022 there were 167 days with confirmed SRKW presence in the Salish Sea, and an additional 63 of days of speculated SRKW presence. Speculated days primarily occurred in Jan-Mar and Oct-Dec, representing both the overall more robust whale tracking that occurs from Apr-Sept and the increased usage of the northern Salish Sea (with lower regional observer effort) in the winter months. Figure 4 shows 2022 speculated days by month compared to average monthly presence from 1998–2002 (from Olson et al., 2018) and 2018–2022 (from this study).

Table 1 Monthly presence of SRKW in the Salish Sea from 2018–2022.

The letters in each cell refer to the pod(s) who were present at least once that month and the number indicates the number of confirmed days of SRKW presence of any pod. The far right column indicates the total number of days present for each year, while the bottom row indicates the average number of days present in a given month across all five years.

	Jan	Feb	Mar	Apr	May	Jun	Jul	Aug	Sep	Oct	Nov	Dec	Total	
2018	JK
4	J
1	JK
7	J
5	NONE	JKL
12	JKL
20	JL
10	JKL
29	JK
7	JK
18	JKL
13	126	
2019	JKL
11	JL
4	J
8	J
11	J
4	NONE	JK
2	JKL
17	JKL
20	JL
17	JKL
21	JKL
10	125	
2020	JKL
8	JK
6	JL
12	J
6	NONE	L
2	JKL
21	NONE	JKL
22	J
13	JKL
22	JK
14	126	
2021	JKL
4	JKL
6	J
13	J
7	NONE	NONE	JKL
5	JL
4	JKL
20	JKL
12	JKL
21	JKL
11	103	
2022	JL
10	J
11	J
15	J
11	J
8	JL
11	JKL
14	JKL
9	JKL
15	J
20	JKL
21	JKL
22	167	
Average	7.4	5.6	11	8	2.4	5	12.4	8	21.2	13.8	20.6	14		

Statistical analysis

A negative binomial model was used to assess the seasonal effect by area for the 2018–2022 data. Using a type III ANOVA to assess the main effects, season, area, and the interaction effect between season and area were all significant (p < 0.001) while year was not significant (p = 0.06). An interaction plot of the estimated marginal means can be seen in Fig. 5. Regardless of area, whale presence was lowest in the spring and winter. The highest predicted whale presence occurs in the CSS in summer, PS in fall, CSS in fall, and CSS in winter.

The generalized additive model utilized to look at seasonal (Apr–Sep) whale presence from 2004–2022 found significant effects of both month (EDF = 3.53, F = 16.03, p < 0.001) and year (EDF = 1.00, F = 50.76, p < 0.001). Partial effects plots show that sightings were lowest in April and May and highest in September. When accounting for between-month variation, spring and summer presence steadily declined in linear fashion across the 2004–2022 time period (Fig. 6).

Figure 2 SRKW whale days by season and sub-region.

Cumulative 2018–2022 sightings of SRKW presence for winter (Jan–Mar), spring (Apr–Jun), summer (Jul–Sep), and fall (Oct–Dec) with data labels indicating number of days for reach region by season. Pink indicates the central Salish Sea (CSS), yellow Puget Sound (PS), and teal the northern Salish Sea (NSS).

Figure 3 Confirmed seasonal SRKW sightings in the Salish Sea from 2018–2022.

Each map shows the initial location of an SRKW sighting, defined as a unique group seen on a unique day. Blue dots indicate winter sightings (Jan–Mar), green spring (Apr–Jun), yellow summer (Jul–Sep), and purple fall (Oct–Dec). Sightings were tracked in all the Salish Sea waters east of Otter Point near Sooke, BC on a daily basis from January 1, 2018 to December 31, 2022. Map created by Tomis Filipovic.

Figure 4 Average monthly SRKW presence from 1998–2002 and 2018–2022 plus 2022 speculated days present.

The dark blue line indicates the average number of days each month SRKWs were confirmed to be present in the Salish Sea from 2018–2022. The yellow line indicates average monthly presence from 2002 per Figs. S3 and S4 in Olson et al., (2018). The pink line shows speculated SRKW presence for 2022 only, defined as confirmed days days where SRKW were present plus days where presence was presumed due to typical travel routes and speeds regardless of visual or acoustic confirmation.

Figure 5 Estimated marginal means interaction plot showing predicted days present by season and sub-region.

NSS, Northern Salish Sea; CSS, Central Salish Sea; PS, Puget Sound. A negative binomial regression model found that season, area, and the interaction between season and area were all significant, while year was not. Bars indicate 95% confidence intervals.

Figure 6 Generalized additive model results for SRKW Salish Sea whale days from April-September for 2004–2022.

Partial effects plots for the smooth terms for month (top) and year (bottom). Circles represent the partial residuals and the shaded area indicates standard error of the partial effect.

Discussion

These results show a continued decline in annual average SRKW presence (129.4 days or 35.3% of the year) and a new record low for annual days present (103 days in 2021). Previously published data from 1976–2014 reported an annual average SRKW presence of 193.1 days (or 52.9% of the year) across the 39 year time period with a low of 139 whale days in 1977 (Olson et al., 2018). Specifically, in previous decades, 71.7% of Central Salish Sea days occurred between May and September, the months formerly considered the core summer season for SRKW in the Salish Sea (Olson et al., 2018). Similarly, Hauser et al. (2007) reported SRKW presence an average of 79.3% of the time between May and September from 1996–2001. In this data set, looking at 2018–2022, only 49.2% of CSS whale days occurred during the months of May–September, with the SRKW present an average of just 32.6% of the days during these months in this study. From 2018–2022, the most CSS days occurred in the months of September (94, 20.66%), July (60, 13.2%), November (48, 10.6%), and March (45, 9.9%). This indicates that the previously reported declining spring presence of SRKW (Shields, Lindell & Woodruff, 2018) is continuing, and indeed appears to be expanding into the months of July and August, with only September SRKW presence remaining similar to historic numbers.

Olson et al. (2018) also reported what they noted as “anomalies” in their data set, with no SRKW reports in April of 2009 or 2013, the only months on record with no SRKW presence in the Salish Sea from 1976-2014. We have documented the continuation of that trend now expanding to additional months, with no SRKW presence in May of 2018, 2020, or 2021, June of 2019 or 2021, or August of 2020. The lack of SRKWs in the Salish Sea during what used to be the core summer months is becoming less anomalous and now more expected in the modern era. As described in Ettinger et al. (2022) and elsewhere, this shift in habitat usage is linked to changes in prey availability.

A recent assessment of April-October Chinook salmon availability for the SRKW in the Salish Sea and off the west coast of Vancouver Island over the last 40 years found that overall salmon abundance available to the whales has declined, with the models predicting an energetic deficit for the whales in six years across the study period, including 2018, 2019, and 2020, the final year for which analysis occurred (Couture et al., 2022). The relative contribution of different Chinook salmon stocks and their availability to SRKW has been changing, with Columbia River stocks specifically increasing in their importance to SRKWs compared to Puget Sound stocks. This may help explain the continued decline in presence of SRKW in what was historically their core summer habitat in the Salish Sea.

Technical committees of the Pacific Salmon Commission provide annual reports on indicator stocks for all naturally spawning salmon populations within US and Canadian waters. For Fraser River Chinook salmon, the historic source of most SRKW prey during the spring and summer months in the Salish Sea (Hanson et al., 2010), much of the data comes from escapement monitoring for five indicator stocks that return through the Salish Sea from April to September: Fraser Spring-Run 1.2, Fraser Spring-Run 1.3, Fraser Summer-Run 1.3, Fraser Summer-Run 0.3, and the Harrison River (Fall-Run 0.3). Escapement is calculated via visual spawning surveys which are generally biased low but are considered accurate for determining overall trends (Tompkins & Baxter, 2015).

The 2021 Chinook Technical Committee report (Pacific Salmon Commission, 2021) states that there have been five years (2017–2021) of below-average escapement for Spring 1.2, Spring 1.3, and Summer 1.3 stocks, though improvement was seen in 2020 and 2021. The Spring 1.3 and Summer 1.3 stocks are considered of high conservation concern due to their substantial recent declines, with 2019 being the lowest escapement in 44 years of monitoring. Spring 1.2 are Fall 0.3 are also of conservation concern due to low overall returns, while Summer 0.3 has been increasing and is considered healthy. This study thus demonstrates that the SRKW occurrence shifts in response to Fraser River Chinook availability described in Ettinger et al. (2022) for the years 2001–2017 have continued from 2018–2022.

Olson et al. (2018) had noted that after the 1999–2000 winter, Ks and Ls have increased the number of months they are detected in the Salish Sea by staying in the Salish Sea later into the fall and early winter, a trend that has also continued in the current data. Ks and/or Ls were not present in any January or February from 1978–1999, while they were present for 62.5% of the Januarys and Februarys from 2000–2014 and 2018–2022. Just as the declining spring and summer presence of SRKW correlates to reduced returns of Fraser River Chinook, this increased SRKW presence in Puget Sound in the fall months corresponds to an increased abundance of both wild and hatchery-raised fall and winter chum salmon in Puget Sound over the 1970–2015 time period (Losee, Kendall & Dufualt, 2019).

The seasonal trend of SRKW presence has essentially reversed from 20 years ago (Fig. 5). They used to be present the most from May to September, while May to August is now when they are here the least. Not only has summer presence declined, but winter presence has increased from October to February. Only the transition months of April and September remain similar to historic numbers. The historic context of SRKW habitat usage is important to keep in mind, especially when identifying critical habitat and aiming for population recovery, but given these new data, it is equally important to consider how the SRKW are currently utilizing their habitat, especially when implementing interim management measures to aid in their recovery.

While considering speculated SRKW days present in 2022 raises the annual total SRKW days present from 167 to 230, which is above the historic average annual presence of 193 days per year, we believe the current data still represents an overall decrease in SRKW presence in the Salish Sea rather than a decrease of confirmed SRKW detections. For 2022, confirmed and speculated days are nearly the same during the months of April-September when sightings effort is highest. Most of the speculated days occur in October-March and are primarily due to J-Pod being undetected in the northern Strait of Georgia or inbound/outbound in the Strait of Juan de Fuca. Undetected days of SRKW presence undoubtedly occurred in the historic data set as well, likely also more so in the winter months due to the same decrease in sightings effort and the habit of the whales to spend more time in less inhabited areas during that time of year. The fact that confirmed and speculated sightings are near-identical during the summer months combined with the fact that confirmed winter sightings are higher overall in 2018–2022 compared to 1998-2002 gives confidence that the rate of confirmed SRKW detections remains the same, and thus the declines in abundance reported in this study are an accurate representation of overall trends. Speculated days are included here to give a sense of how important the Salish Sea has become during what used to be considered the “off-season” months of October to March.

As policy makers continue to implement and revise management efforts to support the recovery of the SRKW, it is critical that actions incorporating seasonal and/or spatial components take into consideration recent habitat usage. Both the United States and Canada have implemented protection measures related to reducing vessel disturbance and/or limiting fishing to improve prey availability for the whales based primarily on older presence/absence trends. These types of actions are only biologically meaningful if they overlap in space and time with where the whales are, so studies like these providing periodic updates on shifting habitat usage are essential to informing these efforts and should continue at least every ∼5 years to reflect ongoing changes.

Conclusions

This study demonstrates that SRKW seasonal and annual habitat usage is continuing to shift from historic trends. SRKW presence has declined considerably during most of what used to be the core summer season of May through September, and has increased during the late fall and winter months. These changes correspond to identified shifts in prey availability, with the continued decline of spring and summer Fraser River Chinook, the increasing importance of Columbia River Chinook in the diet of the SRKW (which would take them outside the Salish Sea), and the overall increased abundance of fall and winter chum in Puget Sound. When managing critical habitat and implementing other area-based policy measures to aid in the recovery of the endangered SRKW, it is important to consider both the historic and current habitat usage, which reflect differences in prey availability. Historic trends must be taken into consideration for long-term habitat protections, while current presence/absence is key to consider for short-term/immediate protection measures such as vessel exclusion zones or area-based fisheries closures. With changes in habitat usage occurring both year to year and from decade to decade, it is reasonable to expect that SRKW presence in the Salish Sea will continue to vary as prey stocks recover or decline and it is important to continue this type of annual monitoring to document these changing patterns. For now, the SRKW have greatly reduced their Salish Sea presence in the spring and summer, with fall and winter now being the seasons they are more likely to occur.

Supplemental Information

Data S1 Raw data for Figures 2–5 and Table 1

Click here for additional data file.

Long-term monitoring studies of highly mobile species such as the SRKW over large geographic areas such as the Salish Sea are only possible with significant community effort. The Pacific Whale Watch Association and Orca Network were key partners in this regard. We also thank the hundreds of individual whale spotters that contributed reports, photographs, videos, and audio clips that contributed to the ability to complete this study. Extra appreciation is given to the following individuals for their long-term support of this project, help in tracking down media to confirm reports, and/or identification assistance: Alisa Lemire Brooks, Erin Gless, Rachelle Hayden, Johannes Kreiger, and Sara Hysong-Shimazu. Michael Weiss assisted with the development of the GAM. This manuscript was greatly improved thanks to feedback from Claudio Vasapolo, Maria Garagouni and Julia Saltzman.

Additional Information and Declarations

Competing Interests

Author Contributions

Data Availability

The author declares that there are no competing interests.

Monika W. Shields conceived and designed the experiments, performed the experiments, analyzed the data, prepared figures and/or tables, authored or reviewed drafts of the article, and approved the final draft.

The following information was supplied regarding data availability:

The raw data is available in the Supplemental File.

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
