# Peer review of "–2022 Southern Resident killer whale presence in the Salish Sea: continued shifts in habitat usage"

_PeerJ, doi:10.7717/peerj.15635_

## Round 0.1 · original submission · Major Revisions

Dear Author, thank you very much for submitting to PeerJ. Your manuscript is particularly interesting and the reviewers and I agree that it should be published. Notwithstanding, some critical issues emerged from the reading, as suggested by the reviewers. I agree with their comments, above all the fact that the analysis you did should be completed with a more “sophisticated” analysis considering some environmental variables characterising the areas under study, apart the prey availability. This could give a clearer vision of the presence of orcas and how they distribute both spatially and temporally.

·

Basic reporting

Writing is clear and unambiguous throughout the text. Have noted a few typos and sentences that could be restructured for formality’s sake in the general comments.
Figures are relevant and informative, mostly of sufficient quality, with good captions. I have added comments where there is room for improvement:
1) Figure 1: The lines and arrows are a tad confusing. Perhaps colour the study area border line differently to the others and place names closer to respective arrows? “study area border” would not be necessary if you move “CSS” closer to that arrow. Alternatively, remove the arrows entirely and just put the three place names in the middle of each sub-area. Also, while the target readership for this paper will likely already be familiar with the study area, you should definitely include an inset map of the greater region. Further, please indicate some of the locations named frequently in the text, such as the straights of Juan de Fuca, Otter Point, Sooke etc.
2) Figure 2 would be much better as a bar chart, which are safer for the human mind to interpret than pie charts, as well as easier to compare to each other. 2A would be more informative as a bar graph (panel A) with the number of sightings rather than the percentage for each season. Then B, C, D, E can be merged into a single bar graph (panel B), where each seasonal bar is split by colour into the percentage of sightings per area, either stacked or dodged.
3) Figure 5: caption states that yellow line is from 2002 only, legend says it’s the five-year average.

Experimental design

The methods are sufficiently clear and can easily be replicated with the raw data provided, or on any similar dataset. However, the actual statistical analysis (t-tests) is first mentioned in the Results, when it should be described in the last paragraph of the Methods section.

I have two main concerns with the otherwise very thorough study. My first question is about the comparison to a five-year period from twenty years ago. In the abstract, you explicitly mention data from 1976 to 2014, but the actual comparison in the results is made to data from 1998 to 2002. It is very important to compare to historical patterns, but it is not made sufficiently clear in the text why that specific period was chosen. Is it merely because that is the only comparable available dataset? If so, that should be elaborated on in the Methods section. If that is not the case, then it would be more informative to look at temporal trends using some form of time series analysis (e.g., https://fromthebottomoftheheap.net/2011/06/12/additive-modelling-and-the-hadcrut3v-global-mean-temperature-series/, https://fromthebottomoftheheap.net/2014/05/09/modelling-seasonal-data-with-gam/). That way, you can identify when exactly the changes presented here actually started happening, whether they followed some specific event or can directly be linked to trends in salmon abundance etc. In fact, even another simple comparison to a more recent time window would be useful, e.g., if you have data from 2008–2012, you could compare twenty years ago to ten years ago and to today. If ten years ago already looked similar to today’s patterns, then the changes started occurring prior to that, whereas if it looked more like twenty years ago, then the changes are more recent.

My second query is to do with spatial, as opposed to seasonal, changes in habitat use. I would like to see a more detailed analysis comparing the three sub-areas of the study area at least within the current time window, if not also compared to twenty years ago or more recent years. As your paper highlights the importance of understanding core habitat use, you should also present any changes (or in fact, lack of changes) within that core habitat. If any of the three areas is being used more or less than previously, that indicates fine-scale changes within the habitat, which local stakeholders should be informed of. If they are all being used for comparable fractions of the year, that indicates that the issue is broader than the immediate study area and highlights the need for wider protective measures. You can test this with something simple, such as a linear model (or GLM or GAM) of sightings ~ area:year.

Validity of the findings

No comment

Additional comments

The study is simple yet robust, providing a good overview of current habitat use patterns and some insights into how this has changed over time for the SRKW population. The current dataset spans a five-year time period, with very good coverage of each season within those years, so any conclusions can be safely said to represent a real pattern rather than a fluke. As I mentioned above, if the data allow it, I would highly recommend doing some more in depth analysis of spatio-temporal trends.

Thank you for explicitly mentioning observer bias and how it was accounted for. Do you know if there were differences in observer effort due to covid restrictions during 2020–2021? It would be worth directly mentioning even if no differences were noted, as many citizen science projects around the world were affected by the pandemic and it is likely your readers will be curious about it.

Line 132: should say “increasing popularity”
Line 154: should say “et al.”
Line 170: worth pointing out that the “most sightings” referred to regarding Puget Sound comprised just over 51% of the total fall sightings, as opposed to the previous percentages mentioned. That is, the fall sightings were less concentrated in a given area, but were the other 49% equally spread across both the other areas? This ties in with the need for a more detailed spatial analysis of the sightings.
Line 236-237: Rephrase a bit more formally?
Line 331: missing “of”
Line 340: species name should be in italics

·

Basic reporting

• The author used clear English throughout the manuscript. I made a couple of comments about minor grammatical errors throughout via comments on the PDF of the manuscript.
• The background is brief, but mostly sufficient. If the author feels it could benefit the manuscript, I think the introduction could start a little bit broader, discussing Orcas and Pod behavior in general. I think the manuscript would be more accessible to a wider audience if a bit more background on pod identification were provided.
• The structure of the article is professional, and the raw data is shared.
• While there is a not an explicit hypothesis stated, the study does a very good job of meeting its aim of “The aim of this study is to provide an update on the seasonal and annual usage of the Salish Sea by each of the Southern Resident pods and the SRKW population as a whole by assessing confirmed presence over the five-year period from 2018-2022.”
• I would love the see a figure which is a time series of how whale encounters change each year, just mean number or percent of whale days each year.
• Table 1: I like this table a lot. However, I think the blue scale is a bit hard to see. It could be hard to see if this paper were printed out. It is up to you, but I feel that using another pallet where each color corresponds to one, two, or three pods present. The color pallets here are color blind friendly and could be of use to you: https://cran.r-project.org/web/packages/viridis/vignettes/intro-to-viridis.html
• Figure 1: I find this map a bit hard to follow. Is it possible to add coordinates, a scale bar, and North arrow? I also typically feel like it is helpful for readers to visualize the location of the study in relationship to a “zoomed out” map. Figure 1 of this paper: https://sci-hub.ru/https://conbio.onlinelibrary.wiley.com/doi/abs/10.1111/cobi.12478 does a good job doing this with a small island off the coast of Costa Rica.
• Figure 2: I usually do not think that Pie Charts are appropriate for ecological data. However, in this case since you are displaying proportions, it is appropriate. If possible, can you add the number of sightings? In C and D rather than putting PS next it’s section can you make the text darker or add a line to this “slice”?
• Figure 3: Can you switch the scale bar to kilometers or add another scale bar with kilometers? Can you label the maps panels A, B, C, and D and add this to the caption. Additionally, the blue is a bit hard to see on the blue water, can you make it a darker blue or add more contrast?
• Figure 4: It is unclear to me why the data from 2002-2018 is not included in this graph. Other than that, I like the figure, and I like how statistical significance is indicated.
• Figure 5: Is there a reason for not including the 2018-2021 data in this graph? Just something to think about.

Experimental design

• The research fits within the aim and scope of the journal. I think this is a good fit for the work, because while the findings are not “novel” they fill an important niche by providing an update to the existing literature.
• To truly make the work replicable, it would be great to specifically state the statistical platform they used, and if applicable provide the code.
• Was there a permit required for this work? If so, I feel that it would be worth mentioning this in the methods.
• I think it would be helpful for the author to provide an explanation for how or if survey effort varied under different environmental conditions.
• I am not sure what platform was used for statistical analysis or making figures. This should be stated in the methods. I would prefer for the methods to be split into categories for whale sightings, statistical analysis, and pod identification. Or something along those lines.

Validity of the findings

• While the author did not use the most complex form of statistical analysis, in this case, they were able to display the aim of their project with the stats they used.
• I think the authors do a good job of assessing the limitations of the study; however, I think they do not address how environmental variation could contribute to their observed trends. While they spend a good time talking about prey availability, I think talking about environmental variation is just as important. Even if they only discuss it in the context of not including environmental variation as a limit of the study.

Additional comments

I have provided in text feedback, in addition to the feedback here. I think this manuscript overall is very good. I have made suggestions on how the author can better assess the study limitations, incorporate ecological literature, and improve figures in the comments above and in the text.

---

## Round 0.2 · accepted · Accept

Dear author, as you can read the reviewer agreed with your changes, and also I am satisfied with the answers and changes given. You addressed all the issues and I think the manuscript greatly improved. Reviewer noted some typos, so please, address these last comments and then the manuscript will be ready to be published.

·

Basic reporting

The manuscript has been hugely improved, with very informative context added to introduction and discussion. I thoroughly approve of the addition to the title, it makes it more impactful than the original.

I have some very minor editorial notes on the text itself:

Replace hyphens in number/date ranges with en-dashes throughout manuscript.
Line 36: missing “of” after “wide variety”.
Line 39: should say “many”, not “may”.
Table 1. Colours listed in the legend do not match table colours. Make sure to update.
Figure 2. Legend should say “for each region”
Figure 3. Legend should say “sightings in the Salish Sea”
Figure 4. Should the legend say “average monthly presence from 1998 to 2002”? Or is it just 2002 data (in which case not an average)? Make sure the text in line 277-278 matches this.

Experimental design

Author has incorporated reviewer feedback in the methods, resulting in a much more detailed approach and understanding of the SRKW habitat use. The model structure described is appropriate, and all the necessary justifications/explanations are given. The study has thus been elevated from "robust" to "robust and rigorous".

Validity of the findings

Brilliant, no comment.

Additional comments

I am delighted with this version of the manuscript and thoroughly enjoyed reviewing it. The additional information on prey availability ties in very well with the temporal analysis. I believe the study now provides all the necessary spatio-temporal information to facilitate changes in protective regulations in and around the study area.